# Effect of Exposure Concentration and Growth Conditions on the Association of Cerium Oxide Nanoparticles with Green Algae

**DOI:** 10.3390/nano13172468

**Published:** 2023-09-01

**Authors:** Aiga Mackevica, Lyndsey Hendriks, Olga Meili-Borovinskaya, Anders Baun, Lars Michael Skjolding

**Affiliations:** 1Department of Environmental and Resource Technology, Technical University of Denmark, Building 115, DK-2800 Kgs. Lyngby, Denmark; aima@env.dtu.dk (A.M.); abau@dtu.dk (A.B.); 2TOFWERK, Schorenstrasse 39, 3645 Thun, Switzerland; lyndsey.hendriks@tofwerk.com (L.H.); borovinskaya@tofwerk.com (O.M.-B.)

**Keywords:** cellular exudates, uptake mechanism, cellular uptake, nanomaterials

## Abstract

The increasing release of engineered nanoparticles (NPs) into aquatic ecosystems makes it crucial to understand the interactions of NPs with aquatic organisms, such as algae. In this study, the association of CeO_2_ NPs with unicellular algae (*Raphidocelis subcapitata*) and changes to the cellular elemental profile were investigated using three exposure concentrations (1, 50, and 1000 µg CeO_2_/L) at two different algal growth conditions—exponential and inhibited growth (1% glutaraldehyde). After a 24 h-exposure, algal suspensions were settled by gravity and CeO_2_-NP/algae association was analyzed by single-cell inductively coupled plasma quadrupole mass spectrometry (sc-ICP-QMS) and ICP time-of-flight MS (sc-ICP-TOFMS). Concurrent detection of the cellular fingerprint with cerium indicated NP association with algae (adsorption/uptake) and changes in the cellular elemental profiles. Less than 5% of cells were associated with NPs when exposed to 1 µg/L. For 50 µg/L exposures in growing and inhibited cell treatments, 4% and 16% of cells were associated with CeO_2_ NPs, respectively. ICP-TOFMS analysis made it possible to exclude cellular exudates associated with CeO_2_ NPs due to the cellular fingerprint. Growing and inhibited cells had different elemental profile changes following exposure to CeO_2_ NPs—e.g., growing cells had higher Mg and lower P contents independent of CeO_2_ concentration compared to inhibited cells.

## 1. Introduction

Thorough understanding of interactions in the nano–bio interface is a key component in understanding the ecotoxicological effects and bioaccumulation behavior of nanoparticles (NPs) in the environment. Transport through the food web is one of the aspects that has been widely discussed in the scientific literature, but there is currently a high degree of uncertainty associated with bioaccumulation studies of NPs while conclusive evidence of trophic transfer is still lacking [1,2,3].

For trophic transfer, algae are of particular importance since they may be “point of entry” of NPs into the food web [2]. Previous studies have documented the association of NPs to primary producers, such as algae [4,5,6]. This suggests that NPs are able to enter aquatic food webs and potentially become transferred from primary producers to primary consumers and further up in the food webs [7,8]. For bioaccumulation studies with algae, it is however important to consider that at different growth stages algae produce extracellular polymeric substances (EPS) for different functions, e.g., cohesion of biofilm and cell aggregation, water retention, and antioxidant [9]. EPS production can vary as a function of different environmental stressors such as changes in salinity and nutrient availability but also from strain to strain. Multiple publications have shown that NP–EPS interaction occurs though it remains unclear whether NP exposure induces EPS production, e.g., as a stress response or whether the EPS production mediates shedding of NPs from the algal cell. Nevertheless, EPS is believed to play a vital role in the bioaccumulation behavior and toxicity of NPs, hence also in their potential for trophic transfer [10,11].

Only a handful of studies have addressed uptake and/or adsorption of NPs to algal cells, most in a qualitative rather than quantitative manner. Several analytical approaches have been used for this purpose, including transmission electron microscopy (TEM) or scanning electron microscopy (SEM) coupled with energy-dispersive X-ray spectroscopy (EDX) [12,13], confocal fluorescence imaging [14], Raman spectroscopy [15], and NanoSIMS [16]. Recent developments in inductively coupled plasma mass spectrometry (ICP-MS) techniques have enhanced the possibilities for detection and quantification of NPs in single cells. The conventional approach has been to separate the cells from the exposure medium and acid-digest the collected cells to obtain quantitative information on NP association with algae. However, this approach does not provide information on NP adsorption or uptake on a single cell level. The new approach is to utilize single-cell inductively coupled plasma quadrupole mass spectrometry (sc-ICP-QMS) and ICP time-of-flight MS (sc-ICP-TOFMS) combined with dedicated sample introduction systems which allow the measurement of intact cells. While sc-ICP-QMS is limited to the identification of only one element per cell or two at best with fast scanning systems [17], sc-ICP-TOF-MS allows simultaneous detection of entire ranges of elements. Sc-ICP-TOFMS can thereby be used to determine the elemental “algal cell fingerprint” [18]. A recent study by Hendriks and Skjolding [19] described the principles of these techniques, as well as their advantages and limitations when it comes to assessing nanoparticle association with single cells.

Analysis on a single cell level using sc-ICP-MS is being increasingly used for toxicological and ecotoxicological research, either through monitoring cellular elemental content or tracking NP adsorption and/or uptake to cells. Several toxicology studies have investigated NP uptake, e.g., by measuring Ag NP contents in human monocyte THP-1 cell line [20] or by quantitative analysis of Ag NPs and TiO_2_ in lysed mouse neuroblastoma cells [21]. This technique has also been used for analyzing Au NP uptake in the algal species *Cyptomonas ovate* [4] and *Raphidocelis subcapitata* [6], BaSO_4_ NP uptake in *Raphidocelis subcapitata,* and Mg content in *Chlorella vulgaris* cells [22]. Von der Au et al. [23] used sc-ICP-TOFMS to investigate the cell elemental profile of the diatom species *Cyclotella meneghiniana* simulating environmental stress after exposure to different metals. Additionally, sc-ICP-MS studies found association of between 1 to 10,000 Au NPs per algal cell for different forms of Au NPs (10–100 nm), rods (ranging from width 10–70 nm and length 45–300 nm) and wires (width 75 nm, length 300 to 6000 nm) underlying the potential of trophic transfer of NPs from algae to higher trophic levels, e.g., crustaceans [11]. While Au NPs are ideal for proof-of-concept studies because of low toxicity, high stability, low background concentrations, and low detection limits both by chemical quantification methods and by TEM, they have low relevance in relation to environmental impact due to a relatively limited range of applications and low exposures to the environment [24]. In contrast to this, CeO_2_ NPs are used in a wide range of applications ranging from corrosion protection [24], solar cells [25], fuel oxidation catalysis [26], and car exhaust treatment [27] where direct environmental release is to be expected. PVP coated CeO_2_ NPs with a diameter of 5–10 nm have been identified to internalize in algal species (*C. reinhardtii*) and cause membrane damage through direct contact [28] while larger agglomerates of 140 nm did not show internalization [29]. Internalization is suggested to be size dependent while effects such as oxidative stress responses have been found for different sizes and coatings of CeO_2_ NPs in green algae mediated by direct contact [28,30,31]. However, studies have shown potential for green algae to recover from CeO_2_ NPs chronic exposures [32], thus suggesting a mechanisms for shedding or mitigating direct attachment. The underlying mechanism and the relation to natural processes such as growth and test design considerations like exposure concentration are unclear.

The aim of this study was to evaluate the effect of exposure concentration (1, 50, and 1000 µg CeO_2_/L) and growth conditions (exponential growth compared to inhibition with 1% glutaraldehyde) on the potential for association of CeO_2_ NPs with unicellular algae (*Raphidocelis subcapitata*) using two single cell ICP-MS methods (ICP-QMS and sc-ICP-TOFMS). We investigated the potential for using a simplified ICP-QMS approach compared to multi-element analysis by sc-ICP-TOFMS for identifying algal cells associated and non-associated with NPs while also assessing the interaction of EPS or natural organic matter associated with NPs.

## 2. Materials and Methods

### 2.1. Materials

Spherical CeO_2_ nanoparticles (NM-212, Fraunhofer IME, Aachen, Germany) with a manufacturer primary particle size of 10–100 nm determined by TEM and a primary crystal size according to Scherrer 33 nm with specific surface area determined by BET, 28 m^2^/g, were suspended in deionized water using the particle dispersion protocol by OECD TG 318 [33] resulting in aggregates with average hydrodynamic diameters of 300 nm with a size distribution from 100–350 nm (PdI 0.38 ± 0.05 and zeta-potential 28 ± 3 mV), determined by dynamic light scattering (DLS) (Zetasizer Nano ZS, Malvern, UK).

### 2.2. Algal Growth Conditions

Unicellular green algae *Raphidocelis subcapitata* were used as the model organism for testing nanoparticle association to algal cells. These are crescent shaped and approximately 10 µm long and 4 µm wide. Algal cultures were kept in ISO 8692 algal medium [34] at 20 °C under constant shaking and illumination (80–100 µE·m^−2^·s^−1^) using the LEVITATT algal testing setup [35]. In short, nine wells each containing five scintillation vials fastened with metal clamps on the side of the wells were illuminated from below with an LED source sandwiched between two acrylic plates. The LED source delivered light with an intensity of 108 ± 10 µE∙m^−2^∙s^−1^ measured with an LI-189 Quantum/Radiometer/Photometer (LI-COR, Cambridge, UK) in the “cool-white” spectrum. The LEVITATT was mounted on an orbital shaker (IKA^®®^ KS 260 basic) at 200 rpm in a temperature-controlled incubator at 23 ± 2 °C. The cell densities in cultures were determined by Coulter counting (Beckman Multisizer^TM^ 3, Indianapolis, IN, USA). To ensure that the cultures used in experiments were in the exponential growth phase, a pre-culture was started 2–3 day prior to testing by adding 10^4^ cells/mL to 20 mL glass vials filled with 4 mL ISO algal medium. These vials were incubated in the LEVITATT test system as described above.

### 2.3. Algal–CeO_2_ Interaction Tests

The test setup for algal exposure to NPs included three different CeO_2_ exposure concentrations: 1, 50, and 1000 µg/L. The range of concentrations was chosen to represent environmentally relevant concentrations (1 and 50 µg/L) and an ecotoxicology test relevant concentration (1000 µg/L). Two different test conditions were used—in one batch cells were allowed to grow exponentially, and in the other batch a growth inhibitor (glutaraldehyde to a final concentration of 1% (*w*/*w*)) was added to stunt algal growth and maintain the cell concentration constant throughout the testing, storage, and sample preparation. Furthermore, the inhibition of growth limits the release of extracellular polymeric substances induced by contact with the CeO_2_ NPs. The samples were prepared in duplicates, one set for sc-ICP-QMS analysis and another set for sc-ICP-TOFMS analysis.

The tests were carried out in 20 mL glass vials. Each vial contained 4.0 mL ISO 8692 algal growth medium (ISO, 2012) and around 10^6^ cells/mL (*R. subcapitata*) added from the pre-culture. As described above the CeO_2_ NP concentrations were 1, 50, and 1000 µg/L. The vials were incubated in the LEVITATT system [35] at 20 °C for 24 h. Additionally, control vials with ISO 8692 algal medium containing 10^6^ cells/mL (*R. subcapitata*) were also incubated. After 24 h, samples were withdrawn from the LEVITATT system and centrifuged at 4000 rpm for 5 min, removing the supernatant and resuspending the pellet with media, this process was repeated three times Algal cell concentrations were determined by a Coulter Counter (Beckman Multisizer^TM^ 3, Indianapolis, IN, USA), and the vials were put in the refrigerator to allow settling of algal cells and to minimize cell divisions after the 24 h exposure test. The algal suspensions were left for 24–96 h to settle and directly prior to analysis the supernatant medium was withdrawn gently by pipette, and deionized water was added to the vials to resuspend the sedimented algae for further analysis.

### 2.4. ICP-MS Instrumentation

The resulting suspensions of algae were analyzed by sc-ICP-QMS (NexION 350D, Perkin Elmer, ON, Canada) and sc-ICP-TOFMS (icpTOF S2, TOFWERK, Thun, Switzerland). All operating conditions can be found in Appendix A [36].

In sc-ICP-QMS analysis ^24^Mg, ^31^P, ^56^Fe were measured for identification of algal cells and their constituents and ^140^Ce to quantify CeO_2_ nanoparticles using the Asperon^TM^ spray chamber and the Perkin Elmer single cell analysis system and single cell application in Syngystix 2.5 software with sample uptake rate of 15 µL/min and 100 µs dwell time. Analysis of ^24^Mg and ^31^P was conducted in standard mode, and for the ^56^Fe analysis dynamic reaction cell (DRC) mode was used with 0.3 mL/min NH_3_ as reaction gas. Each sample was injected three times for a 60 s run.

For sc-ICP-TOFMS analysis, the ICP-TOFMS was equipped with a single-cell sample introduction system (SC-SIS) from Glass Expansion. Samples were injected using a syringe pump with a flow of 10 µL/min and exchanged manually. The instrument was operated in collision/reaction cell mode with 5 mL/min H_2_/He to remove the ArO interference on ^56^Fe. The TOFMS data were acquired with a time resolution of 1 ms and further processed in TOFpilot 2.10 (TOFWERK, Thun, Switzerland) using the liquid reprocessing module. Integrated signals of the analytes of interest were exported as csv files for further data processing. Transport efficiency was estimated using the particle size method with 60 nm Au NPs (NIST, Gaithersburg, MD, USA) and 50 nm Au NPs (nanoComposix, San Diego, CA, USA) for sc-ICP-QMS and sc-ICP-TOFMS, respectively. The recovery of the single cell injection method was determined by comparing the cell counts from Coulter counter measurements to cell counts using the isotopic tracers described above.

## 3. Results and Discussion

For the data interpretation, it is important to underline that both the ICP-QMS and the ICP-TOFMS instruments are able to detect elements present in the algal cells, as well as CeO_2_ nanoparticles in suspension or attached to cells. However, ICP-QMS can measure only one element at a time. Therefore, it is assumed in the data interpretation of ICP-QMS analyses that each signal event of an element present in algae belongs to a single, intact algal cell. ICP-TOFMS, however, simultaneously captures the full elemental fingerprint of a single cell (algal cell fingerprint) as well as CeO_2_ particles associated with the cell. The transport efficiency based on size was 41% and 65% for sc-ICP-QMS and sc-ICP-TOFMS, respectively (Appendix A). The total loss of cells from the centrifugation, settling, and renewal of media was 10.2 ± 6.6%. The number of CeO_2_ NPs in the supernatant after the final media renewal was not markedly different from the media background. The cell count from sc-ICP-QMS compared to the cell count based on Coulter counter measurements were 55 ± 10%.

### 3.1. Selection of Cellular Tracer for R. subcapitata

For identification of algal cells, it is of utmost importance to have sufficient signal above background levels (signal-to-noise ratio). Ideally, the same tracers should be used to identify all cells, independent of their condition; growing and inhibited cells with and without exposure to CeO_2_ NPs. It was documented by Von der Au [23] that the detection of algal cells based on one element alone leads to high false-positive numbers (~30%), and it was anticipated that the use of multiple fingerprint elements would significantly reduce false-positives. The TOF mass analyzer can measure all elements simultaneously thereby providing a clearer picture, whether the detected signals are from cells, EPS, or testing medium. To determine the reliable tracers for sc-ICP-QMS, we compared the number of events detected for each isotope and if this number was within the same range, the elements were included in the tracer selection.

Algal cell counts from sc-ICP-QMS analysis, where the same dilution factors were used, were similar for control cells in both growing and inhibited treatments based on all three elements (Mg, Fe, P). Elemental fingerprint analysis of *R. subcapitata* revealed that P and Fe are the most adequate tracing elements for both sc-ICP-TOFMS and sc-ICP-QMS based on the detection limits and sufficiently high signal-to-noise ratio in both growing and inhibited cells. While Mn seemed to be suited for sc-ICP-TOFMS analysis, the signal-to-noise ratio for sc-ICP-QMS was too low for reliable cell detection. In the sc-ICP-TOFMS analyses (see Figure 1), Mg and K were clearly detectable in growing cells, but their signal intensities were markedly lower in inhibited cells. Consequently, P, Mn, and Fe signals were used as tracers for algal cells in the sc-ICP-TOFMS data analysis and P and Fe for the sc-ICP-QMS analysis. Further data related to the cell tracer selection and reproducibility for the different samples can be found in the SI.

### 3.2. Cellular Elemental Profile for R. subcapitata Exposed to CeO_2_ NPs

No growth inhibition due to the exposure of CeO_2_ NPs was observed for any of the exposure concentrations based on cell counts by Coulter counting. This was supported by cell counts based on Fe and Mg measurements described above (see Appendix A).

Exposed and non-exposed algal cells were screened for ^24^Mg, ^31^P, ^56^Fe by sc-ICP-QMS, and the obtained intensities were then compared for these three elements to see the changes in the cellular elemental profile for different algal growth conditions (growth/no growth) and CeO_2_ NP exposures (see Figure 2a). The frequency distribution of detected particle events was fitted with a Gaussian function, and from this model the median value of elemental intensities in cell signals were obtained and in addition normalized with the median of P signal intensity. Inhibited cells had substantially lower Mg content, as was expected since glutaraldehyde inhibits the production of chlorophyll in the cell. The cellular content of P was considerably higher for non-growing cells compared to the growing cells. Growing and inhibited cells showed different patterns of response to exposure to CeO_2_ nanoparticles—growing cells had higher Mg and lower P contents independent of CeO_2_ concentration, while inhibited cells did not show a clear trend as a result of CeO_2_ exposure for these two elements. It must be noted that for inhibited cells the P content per cell is semi-quantitative since the signal intensity was too high to acquire accurate intensity readings due to the fixation with glutaraldehyde. The Fe content per algal cell slightly increased with higher exposures of CeO_2_ for growing cells.

In parallel, sc-ICP-TOFMS measurements were performed by screening through a wider range of elements present in algal cells. These measurements showed that two distinct ionomic profiles could be recognized between the growing cells and the inhibited cells (see Figure 2b). P, Mn and Fe were observed in both populations; while the contents of Mg and K were clearly decreased; the Fe content slightly increased in the inhibited populations as observed similarly with the sc-ICP-QMS measurements. As mentioned previously, the observed Mg decrease in the inhibited cells can be explained by the inhibition of chlorophyll production by glutaraldehyde, while the decrease in K can also be related to inhibition of primary cellular functions. Likewise, as with the analyte event count, two distinct patterns in terms of elemental fingerprint can be observed for the growing and inhibited cells, respectively, under different exposure conditions.

It is important to note here that although the same elemental profile would be expected independently of the analytical method used, data treatment plays a key role in the produced results. Indeed, for sc-ICP-QMS data, all signals were considered to be cells while for the ICP-TOFMS data only events containing simultaneously signals for P, Mn, and Fe were considered to be cells events. Consequently, various complexes forming in the medium, as well as cellular fragments and EPS would possibly lead to different fingerprints allowing for exclusion of these components.

### 3.3. CeO_2_ Nanoparticle Association with Algal Cells

Figure 3 depicts the percentage of algal cells associated with CeO_2_ NPs in comparison to the total number of cells detected in the different samples. Although the control samples were not expected to contain any CeO_2_ NPs, this was only true for the first measurements, i.e., before any of the exposed samples were measured. Indeed, even though the system was rinsed to flush out possible contamination after the exposed samples, there was some carry-over of CeO_2_ NPs to the subsequent control measurements on the following day. CeO_2_ association levels were still very low namely <1% for the growing control cells and 2% for the inhibited control cells.

At 1 µg/L CeO_2_ the number of particles detected was very low and close to background concentrations. The exposure concentration may therefore have been too low to quantify CeO_2_ association with cells. However, at this concentration both growing and inhibited cells showed that only 5% or less of the cells in the algal population were associated with CeO_2_ particles. When considering the total amount of CeO_2_ NPs present in the suspension at 1 µg/L, around half was found to be associated with the cellular fingerprint signal. However, the number of detected CeO_2_ particles was very low, making it difficult to illustrate a trend. For 50 µg/L CeO_2_ exposure, non-growing cells were more likely to be associated with CeO_2_ NPs than growing cells (16% and 4%, respectively). However, most of the CeO_2_ NPs present in the suspension were unassociated CeO_2_ NPs, i.e., more than 90% were freely dispersed CeO_2_ NPs or CeO_2_ NPs associated with other organic complexes (e.g., EPS) that did not respond to the cellular fingerprint for both growing and non-growing cells. Similarly, for the highest CeO_2_ NP exposure concentration (1000 µg/L) nearly all the CeO_2_ was found not to be associated with algal cells, but since the number of CeO_2_ NPs was much higher than the number of algal cells, 28% of growing and 85% of non-growing algal cells were found to be associated with CeO_2_ NPs. Thus, care should be taken when extrapolating results obtained for relative high exposure concentrations of NPs (>1000 µg/L) as the onset of the observed response could be driven by physical phenomena that cannot be directly extrapolated to environmentally relevant concentrations, as is the custom in traditional risk assessment.

It should be noted that although concurrent signals have been assumed so far to represent cell-NP association, it is possible that these signals originate from an independent CeO_2_ NP and an independent algal cell reaching the plasma at the same time. Consequently, the CeO_2_ NP and the algal cell will be considered as associated when in fact they are not. While it is not possible to determine a posteriori what happened, the probability of such a case can be determined by concurrency analysis and is strongly influenced by the particle number concentration (PNC) in the sample, here high CeO_2_ NPs PNC. Ideally, the 1000 µg/L exposure of the CeO_2_ NP sample should have been diluted further with respect to the CeO_2_ NPs, but this would simultaneously have led to a lower concentration of the algal cells which was undesirable. To check for coincidental nanoparticle and algal cell events, a simple concurrency analysis was performed. Briefly, the probability of two independent signals captured at the same time is the product of the two independent probabilities of measuring a single analyte at a given time point, P(element_1_ ∩ element_2_). Since it was already established that Mn and P are both considered as signals belonging to the same cell, the probability of Mn and Ce coincidentally being captured at the same time was calculated and yielded <0.01% concurrent events (Appendix A). Hence, although there is a possibility that the assumed CeO_2_ association with algae is a false positive, it is still quite low even for the highest exposure concentration. Because of the relatively wide size distribution of the CeO_2_ NPs (100–350 nm), it was not possible to determine whether the cells were associated with one or multiple NPs based on the relative signal intensity.

The coincident occurrence of cell signals with ceria particles is assumed to be evidence of CeO_2_ nanoparticle association with algal cells (either adsorption or uptake of nanoparticles to algae cells). While the incidence of cell fingerprint elements with Ce signals could be observed in the sc-ICP-TOFMS data, this was not possible with sc-ICP-QMS data. Hence, the working assumption for QMS data was that after washing the samples multiple times, all unbound CeO_2_ particles were removed and the remaining CeO_2_ particles must be bound with cells. However, with the cell washing procedure applied here, it was obvious that the sample preparation did not separate cells from free CeO_2_ in the medium, since there were more Ce events detected than cell fingerprint events (see Appendix A). This observation indicates that extra washing steps or other separation techniques were necessary, especially for the higher CeO_2_ concentrations.

In this study, the results obtained through use of these two instruments allowed the relative abundancies to be revealed of various cellular fingerprint elements in cells exposed to CeO_2_ NPs. For cellular association with CeO_2_ only sc-ICP-TOFMS data were presented, since with this type of sample preparation sc-ICP-QMS cannot provide reliable data for cell association with CeO_2_ nanoparticles. Indeed, in sc-ICP-QMS data, algal fingerprint and Ce signals cannot be measured simultaneously for the same injection and potentially lead to false positive measurements of CeO_2_ NPs not associated with algal cells.

Previous studies using small CeO_2_ NPs with a primary diameter of 5–10 nm showed internalization [28], however for larger CeO_2_ NPs (140 nm) [29] internalization was generally not observed. Similarly, the present study did not specifically observe internalization, as the difference between the growing and inhibited exposures show a markedly different trend, of higher association of CeO_2_ NPs with the inhibited exposures, suggesting a rapid clearance mechanism or no internalization in the growing exposure, but relatively high degree of adsorption in the inhibited exposures, especially at 1000 µg/L (Figure 3).

A proposed clearance mechanism for algae could be the release of extracellular polymeric substances or by cell division. In order to limit the effect of these mechanisms algal cells were inhibited with glutaraldehyde (1%) to stunt growth and metabolic functions. In Figure 3, a higher association with CeO_2_ NPs can be observed for the inhibited cells, which supports the hypothesis that growth or EPS is a mitigating factor of algal association with nanoparticles. This is important since the extrapolation of results obtained with non-growing algal cultures may overestimate the potential for trophic transfer of CeO_2_ nanoparticles in the environment where growth would be a natural part of the cellular cycle. Another study also showed algal interaction of Au NPs ranging from 10 to 100 nm having an association in relation to the exposure concentration ranging from 0.21% to 0.01% respectively [11]. Low accumulation potential was also observed for Ag NPs ranging from 0.44 ng/10^5^ cells to 0.68 ng/10^5^ cells, dependent on coating (curcumin > tyrosine > epigallocatechin gallate) [37]. However, due to the choice of method for quantifying the metallic content and separation techniques, no information is available with regard to whether the NPs are associated with living cells or with EPS or other natural organic matter, which can cause misinterpretation of the available fraction for trophic transfer.

The interaction between algal cells (*R. subcapitata*) and twelve different NPs (6 types of TiO_2_, 3 SiO_2_, 1 ZnO, and 2 CeO_2_) was also studied with flow cytometry, and a general increase in the modified granularity as a function of concentration for eight (6 TiO_2_, 1 ZnO, and 1 CeO_2_) out of twelve NPs tested indicating association of NPs with algal cells was found [38]. Similar concentration dependent increase in NP-algal cell association was observed in this study. However, it should be highlighted that the elevated concentrations could induce effects such as entrapment of algae in NP clusters which would most likely not occur at environmentally relevant concentrations. In environmentally relevant settings, substances such as natural organic matter (NOM) would also be present which has been shown to decrease the toxicity of NPs [39,40]. When involving NOM in the test setup, the complexity of assessing the interaction of NPs with algal cells increases. However, utilizing sc-ICP-TOFMS to map multiple elements simultaneously will allow for distinction of algal cells associated with NPs or assess other compounds that will interact with NPs in environmentally relevant samples.

## 4. Conclusions

In general, a very low degree of interaction between algae and CeO_2_ nanoparticles was documented at environmentally relevant concentrations. Less than 5% of cells exposed to 1 µg/L were associated with CeO_2_ nanoparticles in growing and non-growing cultures, and at 50 µg/L only up to 16% of the cells in a non-growing culture and 4% in a growing culture interacted with CeO_2_ nanoparticles. At a higher concentration (1000 µg/L), similar to concentrations used in guideline ecotoxicological testing, much higher association was observed (up to 85%). Our results show that extrapolation of results obtained at high exposure concentration may not be valid for environmentally realistic conditions, since the higher NP concentration may have caused testing artefacts induced by increased NP to cell ratio. Furthermore, the marked difference between the association of nanoparticles and algal cells in growing and non-growing treatments indicate that growing cells may actively shed nanoparticles due to growth or release of EPS, or that non-growing cells might have a compromised cell wall that is more prone to permeation or adsorption of nanoparticles. This limits the value of results obtained with non- or slow-growing algal cultures with respect to extrapolation to environmentally realistic conditions.

## Figures and Tables

**Figure 1 nanomaterials-13-02468-f001:**
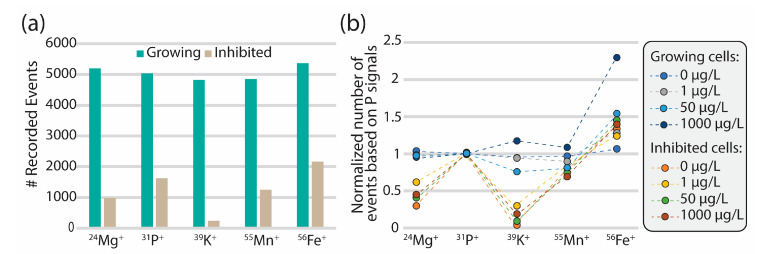
Result of sc-ICP-TOFMS analysis of elements in exponentially growing and growth inhibited algae (*R. subcapitata*). (**a**) Overview of the number of events detected for ^24^Mg, ^31^P, ^39^K, ^55^Mn, and ^56^Fe for both growing and inhibited control cells by sc-ICP-TOFMS. (**b**) To identify elements suitable as cell markers, the number of events for each analyte was normalized to the number of ^31^P events for each concentration of CeO_2_ NPs (normalization with respect to Mn signals is presented in the SI). Note that the dashed lines between the data points are not supported by any model or data but were added to help guide the eye.

**Figure 2 nanomaterials-13-02468-f002:**
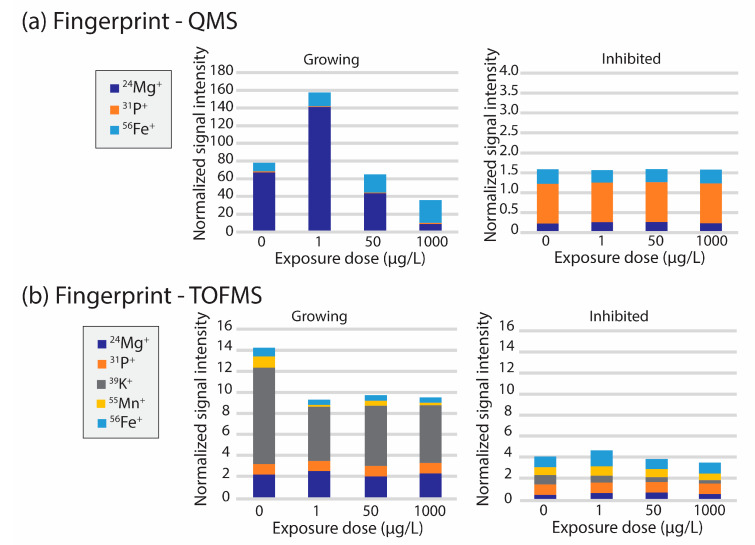
Normalized elemental cellular fingerprints for the freshwater green algae *R. subcapitata* measured by (**a**) sc-ICP-QMS and (**b**) sc-ICP-TOFMS for growing and inhibited populations. Signals were normalized using the median of P signal intensity as a reference. Note that the error bars were removed here to facilitate the visualization of the trends. Figures with error bars can be found in Appendix A in the Supplementary Information.

**Figure 3 nanomaterials-13-02468-f003:**
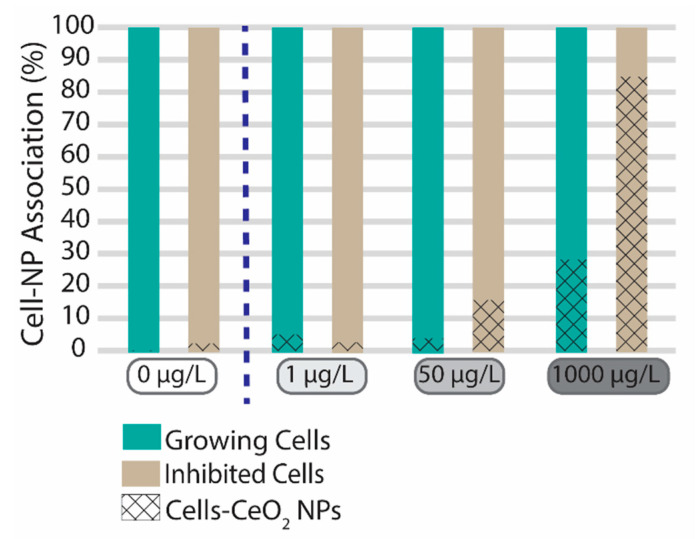
Overview of the cell-CeO_2_ NP association ratio for growing (green) and inhibited (beige) algal cells (*R. subcapitata*) exposed to different concentrations of CeO_2_ NPs based on sc-ICP-TOFMS analysis.

## Data Availability

The data can be shared upon contact with the corresponding author (lams@dtu.dk).

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
