# Peer review of "Effect of Exposure Concentration and Growth Conditions on the Association of Cerium Oxide Nanoparticles with Green Algae"

_nanomaterials, 2023, doi:10.3390/nano13172468_

Round 1

Reviewer 1 Report

This high quality research report deals with the study of the possible interaction of cerium dioxide (ceria) nanoparticles and algae using advanced analytical techniques including single-cell inductively coupled plasma quadrupole mass spectrometry. Nanoscale ceria has become one of the most abundant engineered nanomaterials, and thus this study is an important step in the determination of the environmental impact of engineered nanoparticles. The subject of the paper fits well the scope of Nanomaterials journal.

I have the following comments:

1. The same type of CeO2 NPs was already used in the earlier studies including some biomedical experiments. The results of these experiments could be co-analyzed with those reported in this paper to provide a more holistic view.

2. It is still unclear whether CeO2 NPs were internalised by the cells or no. Please explain in more detail.

3. Existing papers concerning the interaction of CeO2 with algae should be discussed to reflect the novelty of this experimental report.

Author Response

The authors appreciate and thanks the reviewer for their constructive feedback on the submitted manuscripts and have attached a table with response to each individual questions/feedback put forward by the reviewer.

Reviewer 2 Report

This paper contains a valuable study on analysis of CeO2 particles on algae using two different techniques. This is an innovative study that should be acceptable for publication after one major issue is resolved.

My only major comment is that the authors need to prove that a more thorough separation process is not needed here such as density gradient centrifugation. A 0-hour control should be performed by adding the algae cells and NPs together and then immediately putting them in the refrigerator followed by analysis. Hopefully, such an analysis will prove that minimal particles become associated with the cells during the separation process. If not, density gradient centrifugation is likely needed to exclude interactions during the separation process. The authors can review a paper published in Nanomaterials back in 2016 (doi:10.3390/nano6100181) that describes a method to separate MWCNTs from bacteria cells and protozoa using density gradient centrifugation for guidance.

Minor comments

Lines 118-119 - Can the authors add a sentence or two here describing how the LEVITAT system works instead of directing to another paper?

Figure 2 – Where are the error bars?

The quality of the writing is good.

Author Response

(The authors gave the same response as above.)

Reviewer 3 Report

The manuscript by Mackevica et al. describes the application of inductively coupled plasma mass spectrometry in single cell mode (SC-ICP-MS) to the analysis of unicellular algae (Raphidocelis subcapitata) exposed to CeO2 nanoparticles under growth and inhibited growth conditions. The use of quadrupole and time-of-flight ICP-MS instruments is considered, as well as the selection of metal tracers for cell identification. Additionally, the variations of metal tracers with respect to the CeO2 concentration exposure and growing conditions is also discussed.

The Conclusions section summarizes accurately the achievements of the work, although their relevance should be evaluated by an ecotoxicologist, which is not my case. Regarding the analytical part of the work, the reliability of the results presented is questionable. A number of comments are included below.

- SC-ICP-MS with TOF analyzer is clearly superior to the use of quadrupole for identification of individual cells due to the multi-element capability of the former. Therefore, there are no solid reasons to include the SC-ICP-QMS data in the manuscript. Especially, because the most relevant discussion was focused on the SC-ICP-TOF-MS data. The title of the work should be reconsidered.

- The cellular fingerprinting procedure should have been explained in more detail. By using SC-ICP-TOF- MS, Mg, P, K, Mn and Fe were selected as tracers for identification of particle events corresponding to algae cells. Figure 1 shows that similar counting of cell events was recorded for the four isotopes selected when using SC-ICP-TOF-MS under growth conditions. However, the behavior changed in inhibited conditions or in the presence of different concentrations of CeO2 (figures S1 to S3). Since the detectability of cells is conditioned by the mass per cell detection limit (in fact, the critical value) of the corresponding isotope, the number of cells counted under each set of conditions could vary between isotopes. This topic should have been considered and discussed in depth in the work. Moreover, since cells have been counted by Coulter Counter, counting recoveries with the different isotopes and conditions studied should have been included to evaluate the performance of the proposal.

The authors finally selected P, Mn and Fe as tracers for algal cells for SP-ICP-TOF-MS analysis, but apart from the statement "only signal events containing simultaneously P, Mn and Fe were considered to be cells" (l.249), the specific data treatment is unclear. In any case, the uncertainty associated to data in figures 1, S1, S2 and S3 should have been included.

- The utility of an experiment involving agglomerates of CeO2 nanoparticles of 300 nm is questionable; since their internalization is not very feasible, adsorption should be expected. Moreover, since the separation of algae was done by settling, the fate of the CeO2 agglomerates and their presence in the analyzed samples is unclear. Different controls should have been included in the experiments and cerium data should have been analyzed to know the fate and occurrence of the CeO2 agglomerates along the experiments (although 1000 µg/L is too high for direct SP-ICP-MS, 1 and 50 µg/L involve feasible number concentrations for such large agglomerates under the SC-ICP-TOF-MS conditions used). Moreover, information about the amount of cerium associated to individual cells would have helped to understand the results presented and evaluate their reliability. Multielement time resolved scans could have also been included (Sup. Info.) for the same reasons.

The explanations of figure 3 (l.267-282) are not clear enough:

"...When looking at the CeO2 present in the medium it was found that around half of it was associated with cells as assessed by using the cellular fingerprint signal. However, the number of detected CeO2 particles was very low, making it difficult to illustrate a trend." Which particles are the authors referring to?

"...However, most of CeO2 NPs present in the suspension were unassociated CeO2 NPs, i.e. more than 90% were freely dispersed CeO2 NPs or CeO2 NPs associated with other complexes present in the medium for both growing and non-growing cells..." Which complexes are the authors referring to?

- The complexity of this type of study requires that the conclusions be validated using other techniques, which has not been the case.

- The procedure and standards used for transport efficiency determination (Sup. Info.) should have been included (three procedures are described in the reference quoted). In any case, the transport efficiency of algae particles with a size of 10 µm is much lower, as it has been reported for plastic microparticles.

- Figure 2. The parameter used in the plots (y-axis: normalized signal intensity) should have been include.

- Paragraph 3.2. In my opinion, referring to four elements as the ionome of the algae is an overstatement.

- Aparently,  figure S8 was not discussed in the manuscript.

Author Response

(The authors gave the same response as above.)

Reviewer 4 Report

This is a short, well-written paper that highlights the benefits of SC-ICP-TOF-MS in contrast with SC-ICP-Q-MS for simultaneous monitoring of several elements from every single cell: nothing that we do not know, but the results reported may be of interest for environmental studies. 

What I miss the most in the work is the lack of quantitative information. SC-ICP-MS is a quantitative method, and I think the authors should make an effort in this regard. Hence, we should know the mass they are detecting for each analyte, the level of precision attainable, and the limits of detection that can be achieved with each type of instrument.

Author Response

(The authors gave the same response as above.)

Round 2

Reviewer 2 Report

The authors have adequately responded to most of the reviewer comments.

For stacked bar plots, I appreciate that ease of viewing makes adding error bars untenable. However, only plotting mean values in the absence of error bars is scientifically unsound. There is a significantly different meaning if a result is 50 +- 5 or 50 +- 50. The authors should make additional plots to add to the Supporting Information showing error bars for all data.

Author Response

We appreciate the reviewers comments, and understand the value of the errorbars for scientific consistency and clarity. Thus, we've added the requested figures in supporting information (Figure S9)